# CoVariance-based Causal Debiasing for Entity and Relation Extraction

**Lin Ren**[1], **Yongbin Liu**[1][*], **Yixin Cao**[2], **Chunping Ouyang**[1]

[1]University of South China, [2]Singapore Management University

{homurat.rl, yongbinliu03, caoyixin2011}@gmail.com, ouyangcp@126.com

## Abstract

Joint entity and relation extraction tasks aim to recognize named entities and extract relations simultaneously. Suffering from a variety of data biases, such as data selection bias, and distribution bias (out of distribution, long-tail distribution), serious concerns can be witnessed to threaten the model's transferability, robustness, and generalization. In this work, we address the above problems from a causality perspective. We propose a novel causal framework called **c**ovariance and **v**ariance **op**timization framework (OVO) to optimize feature representations and conduct general debiasing. In particular, the proposed **c**ovariance **op**timizing (COP) minimizes characterizing features' covariance for alleviating the selection and distribution bias and enhances feature representation in the feature space. Furthermore, based on the causal backdoor adjustment, we propose **v**ariance **op**timizing (VOP) separates samples in terms of label information and minimizes the variance of each dimension in the feature vectors of the same class label for mitigating the distribution bias further. By applying it to three strong baselines in two widely used datasets, the results demonstrate the effectiveness and generalization of OVO for joint entity and relation extraction tasks. Furthermore, a fine-grained analysis reveals that OVO possesses the capability to mitigate the impact of long-tail distribution. The code is available at https://github.com/HomuraT/OVO.

## 1 Introduction

Named Entity Recognition (NER, (Ratinov and Roth, 2009)) and Relation Extraction (RE, (Bunescu and Mooney, 2005)) are both the key fundamental tasks of Information Extraction and both significant predecessor tasks for building knowledge graphs that are used widely in daily life (Chen et al., 2021; Ji et al., 2022; Du et al., 2022a). There are two major methods to extract the final result.

[*]Corresponding Author.

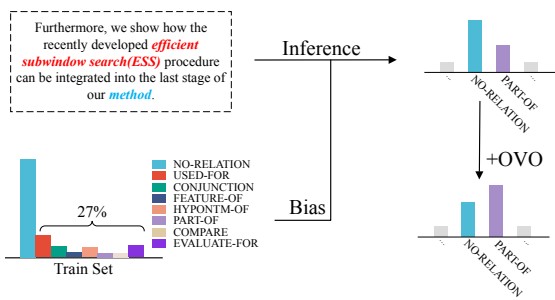

Figure 1: An example of RE on SciERC (the label bias increases the risk of predicting some label possessing majority samples). The confidence at the upper right is predicted by PL-Marker (Ye et al., 2022). The ground-truth relation of the example is **PART-OF**, but due to the bias in the train set, the **NO-RELATION** is inferred. After using our OVO, **PART-OF** is predicted correctly.

One is to build a separate model (pipeline) to get the result in one step (Shang et al., 2022), and the other is to use multiple models (joint) to extract entities and relations in turn (Zhong and Chen, 2021; Ye et al., 2022).

Suffering from various biases in data, such as data selection bias, out-of-distribution, and long-tail distribution (Lin et al., 2022; Wang et al., 2022), models may learn spurious correlations and make erroneous predictions. Although we would like to be able to obtain a uniform (or unbiased) data set, such a dataset is almost nonexistent in real applications. Various biases are always mixed up in real scenarios, for example, selected data in a way that differs from the target population (data selection bias), training and testing data distribution shift (out-of-distribution), the labels with numerous samples may have more chances to inference (long-tail distribution lead to predicting majority bias), and some of the data biases are even unknown. Because statistical correlation is the basis of supervised learning algorithms, the learning algorithms may learn many spurious correlations due to these biases. Finally, this leads to a greatly re-

duced predictive ability of the models. As shown in Fig. 1 (left), there are different degrees of the data bias of long-tail distribution in SciERC (Luan et al., 2018), which is common for other datasets as well, making models focus more on the labels with numerous samples and performing poorly on the others (fewer chances to the labels with few samples).

In recent years, debiasing has become a significant research focus. There are different ways to address the data bias by reweighting samples (Dixon et al., 2018; Rayhan et al., 2017; Wang et al., 2022). These methods focus on the reweights in the sample space and ignore spurious correlations due to data selection bias. Recently, stable learning (Shen et al., 2018, 2020; Zhang et al., 2021) proposed serial ways to de-bias the spurious correlation to find the causal effect of each input feature on the basis of the correlation. Specifically, these methods require extensive calculation resources to identify causality and focus solely on causal effects between features. Additionally, they fail to consider the causal effects for each feature dimension within the same class label.

In this work, we introduce a method called the c**o**variance and **v**ariance **o**ptimization framework (OVO), to optimize the feature vectors from two aspects, decorrelating between features and each feature dimension of the same class label. Our contributions can be summarized as follows:

- From decorrelating between features, inspired by stable learning, we use covariance as the confounder degree of features and try to minimize it in each batch of every epoch to learn the causality between features and targets. We minimize the covariance to reduce the collinearity of each feature, get the causal effects, alleviate the data selection and distribution biases, and enhance feature representation. We call this method covariance optimizing (COP).

- From feature dimensions of the same class label, according to the variance of each dimension, using the intervention of causal inference (Pearl et al., 2016), it can eliminate the confounding bias from the backdoor path by blocking $F \leftarrow L \rightarrow P$ as the causal graph shown in Fig.2, so that model can learn the causality from the causal path $F \rightarrow P$. We explore the causal correlation between the fea-

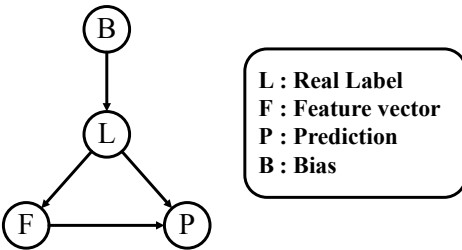

Figure 2: The causal graph of the joint entity and relation extraction task in the classification phase. Bias is something that we can not avoid, such as agnostic data selection bias, data labeling error, and unobservable features.

ture dimensions of the same label and the outcome targets and mitigate the long-tail distribution bias further. We call this method variance optimizing (VOP).

- We combine COP and VOP as OVO. To access the effectiveness of OVO, we conduct experiments on the joint entity and relation extraction task. By applying OVO to 3 strong baselines, we observe significant improvements in the performance of this task, which demonstrates the effectiveness and generalization of our proposed method. These results suggest that optimizing feature representation directly is actually an effective way to get improvement and reduce the bias from datasets without model redesigning, and OVO is a feasible framework in this way.

## 2   Related Work

**Stable Learning**   aims to research how to reduce the agnostic data selection bias and model mis-specification problem and make the feature more causal by reweighting samples. CRLR (Shen et al., 2018) proposes an equation to evaluate the degree of confounders called Causal Regularized Logistic Regression and gains the optimal weight by minimizing the confounder. SRDO (Shen et al., 2020) considers the smallest feature of the covariance matrix as the degree of confounder. StableNet (Zhang et al., 2021) adopts stable learning technology into image classification tasks using deep learning models and the results show the strong domain adaptation of the proposed models.

**Span-level NER and RE Models**   are the approaches using the boundaries of spans as a total feature to recognize the entities and the relation

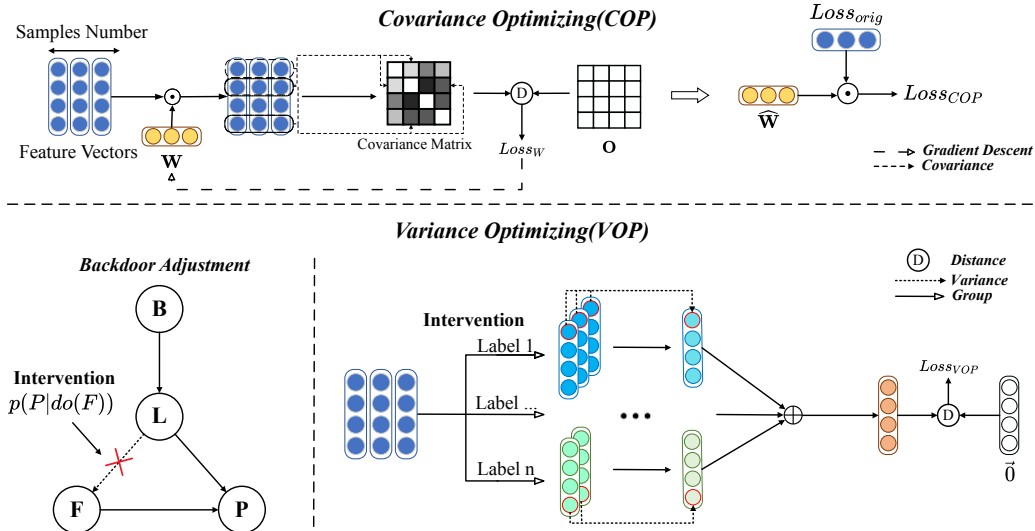

Figure 3: The overview of OVO. In the training phase of the model, we insert COP and VOP as extra optimization for feature vectors. For COP, there is an inner loop to learn the optimal $\mathbf{W}$ using gradient descent, and $Loss_W$ is the loss function of $\mathbf{W}$ calculated according to eq. 4. For VOP, Backdoor Adjustment is used to reduce the bias from datasets. Note that $Loss_W$ only computes the gradient of $\mathbf{W}$, and only the final optimal weights $\widehat{\mathbf{W}}$ created in COP will be applied to the model training process. The $Loss_{VOP}$ will be used in model training directly. $Loss_{orig}$ means the loss employed on the original method; $\mathbf{O}$ means zero matrix.

between two entities. DyGIE++ (Wadden et al., 2019) and SpERT.PL (Sai et al., 2021) extract features of spans and share them between NER and RE tasks. PURE (Zhong and Chen, 2021) uses two separate span models to recognize entities and relations without interaction with each other and proposed levitated markers to reinforce the boundaries information of spans. PL-Marker (Ye et al., 2022) proposes neighborhood-oriented packing and subject-oriented packing methods to leverage the levitated markers.

**Variance and Covariance** are two important metrics in deep learning. In confidence learning (Mukherjee and Awadallah, 2020), they are negatively correlated with the confidence of models and samples. Moreover, in active learning (Cohn et al., 1996), they are correlated with generalization error and bias which can be indirectly reduced by minimizing variance or covariance. Recently, IDEAL (Guo et al., 2022) uses variance as one of the measures for the inconsistency of unlabeled samples. In this work, we use them to estimate the feature representation and correlation between features respectively.

## 3 Method

In this section, we first briefly introduce the general inference process of the joint entity and relation

extraction methods. Then, we will describe covariance optimizing (COP) and variance optimizing (VOP).

### 3.1 Background: Inference Process of Joint Entity and Relation Extraction Methods

In the joint entity and relation extraction task, the goal is to extract both entities and the relations between them from a given text. The general process of this task involves several steps. Firstly, an encoder module is used to transfer pure text to a numerical representation that captures the semantic information. This can be done using techniques like word embeddings or pre-trained language models (PLMs).

$$\begin{aligned} \mathbf{S} &= w_1, w_2, \ldots, w_l \\ \mathbf{H} &= \text{encoder}(\mathbf{S}) \end{aligned} \quad (1)$$

Where $w_i$ indicates the *i-th* word of the given text; $l$ means the length of the text; $\mathbf{H} \in \mathbf{R}^{l \times n}$ represents the representation of the text and $n$ is the dimension size.

For NER, the encoded representations are passed through an entity extraction module to identify and classify entities in the text. The formulations of span-level methods are as follows:

$$\begin{aligned} f_{(i,j)}^{span} &= \text{extractor}(\mathbf{H}, i, j) \\ label_{(i,j)}^{span} &= \text{classifier}(f_{(i,j)}^{span}) \end{aligned} \quad (2)$$

Where $0 \leq i \leq j \leq l$ indicates the boundary of a span; $l$ means the length of the given text; $f^{span}_{(i,j)}$ is the feature of a span; extractor($\cdot$) is a function to gain the feature of given spans; classifier($\cdot$) is a function to predict the label of spans according to its representation.

For RE, $\mathbf{H}$ and the spans of identified entities are used to determine the relations between the entities.

$$
\begin{aligned}
f^{rel}_{(i,j)} &= \text{extractor}(\mathbf{H}, span_i, span_j) \\
label^{rel}_{(i,j)} &= \text{classifier}(f^{rel}_{(i,j)})
\end{aligned} \tag{3}
$$

Where $span_i$ is the entity identified by NER; extractor($\cdot$) is a function to acquire the feature of given relations; classifier($\cdot$) is a function to predict the label of relations. In this work, OVO is used to optimize the $f^{span}_{(i,j)}$ and $f^{rel}_{(i,j)}$ during training.

### 3.2 Covariance Optimizing

When the features exhibit linear correlations with each other, it can lead to instability in parameter estimation. Consequently, the optimal model becomes uncertain, and the presence of selection and distribution biases is amplified during inference. This phenomenon is called *collinearity* (Dormann et al., 2013). For example, $y = x_1 + x_2$ and $y = 100x_1 - 98x_2$ have identical performance if assumption $x_1 = x_2$ holds in train datasets, but if $x_1 = 1.01x_2$ holds in test datasets, the gaps of them will be huge. Besides *collinearity*, the *correlation* between features also poses the same impact in deep learning. One straightforward way to reduce the effect of *collinearity* and *correlation* is to analyze them among features and then remove some relevant features. However, due to the inexplicability and excessive number of features in PLMs, this method is difficult to afford.

The methods of stable learning (Zhang et al., 2021) show that feature representations will be more causal when giving suitable weights for samples in loss calculating during training. Inspired by this, we propose covariance optimizing (COP) to alleviate the *collinearity* and *correlation* between features with low resource consumption, as well as improve the causality of features.

In COP, covariance is used to measure the correlation between features, and the target is to acquire suitable weights by minimizing the covariance among all features using gradient descent:

$$
\widehat{\mathbf{W}} = \underset{\mathbf{W}}{arg min} \sum_{i,j,i \neq j} (cov(\mathbf{W}\mathbf{F}^i, \mathbf{W}\mathbf{F}^j))^2 \tag{4}
$$

Where $\mathbf{F}$ and $\mathbf{W}$ mean the feature vectors and weights; $\mathbf{W} \in \mathbf{R}^n$ is a vector representing all sample weights and $n$ is the number of samples, holding $\sum_i^n w_i = 1$ by softmax function. $\mathbf{F}^i$ means the *i-th* dimension feature of all samples. $cov(\cdot, \cdot)$ is the covariance function.

Next, $\widehat{\mathbf{W}}$ is used to weigh the loss of the corresponding sample.

$$
Loss_{COP} = \sum_{i=0}^{n} \hat{w}_i Loss_{orig}(p_i, t_i) \tag{5}
$$

Where $\hat{w}_i \in \widehat{\mathbf{W}}$ is the final weight for the *i-th* sample; $p_i$ and $t_i$ mean the prediction distribution and real label respectively; $Loss_{orig}$ means the loss employed on the original method.

However, there are two difficulties in performing the above process directly. Firstly, the majority of PLMs have an extensive number of parameters, ranging from hundreds of millions (Devlin et al., 2019) to billions (Du et al., 2022b) and even more, thus the computation of weights for the entire dataset requires significant computational resources, making it expensive and potentially unaffordable during training. Secondly, incorporating features from previous batches or epochs may lead to feature inconsistency due to the continuous evolution of features during training.

To overcome the aforementioned difficulties, inspired by MoCo (He et al., 2020), we only compute optimal weights for samples in the current batch, utilizing previous features and weights as additional fixed information. Furthermore, to ensure feature consistency, we discard the earliest features and weights:

$$
\begin{aligned}
\mathbf{F}_i &= [\mathbf{F}_{i-1}[\lfloor L_{F_i}/(n+1) \rfloor :], \mathbf{F}_{cur}] \\
\mathbf{W}_i &= [\mathbf{W}_{i-1}[\lfloor L_{W_i}/(n+1) \rfloor :], \mathbf{W}_{cur}]
\end{aligned} \tag{6}
$$

Where $\mathbf{F}$ and $\mathbf{W}$ mean the feature vectors and weights; $i$ means the index of the current batch; $L_*$ is the length of $*$; $*_{cur}$ means feature vectors or weights in the current batch; $n$ is the batch number of feature vectors and weights needed to be saved; $\lfloor \cdot \rfloor$ is the floor function; $[l :]$ means to delete first $l$ data.

To further mitigate the feature inconsistency, we fuse and emphasize the current information with previous features and weights. Firstly, we randomly sample from $\mathbf{F}_{cur}$ and the corresponding $\mathbf{W}_{cur}$ until the number of samples is the same as

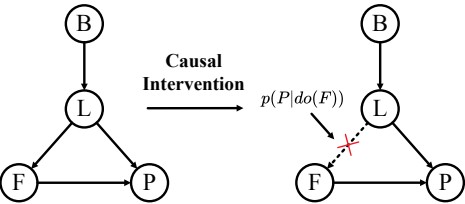

Figure 4: The original causal graph of the joint entity and relation extraction task in the classification phase (left) with its alternate after intervention (right).

$\mathbf{F}_i$.

$$\mathbf{F}'_{cur} = [\mathbf{F}_{cur}[R_1], \ldots, \mathbf{F}_{cur}[R_L]]$$
$$\mathbf{W}'_{cur} = [\mathbf{W}_{cur}[R_1], \ldots, \mathbf{W}_{cur}[R_L]] \quad (7)$$

Where $\mathbf{F}'_{cur}$ and $\mathbf{W}'_{cur}$ are the sets after random sampling; $L$ is the length of $\mathbf{F}_i$, $R_i$ is a random integer from 0 to $L$, and $[i]$ means getting the *i-th* element. And then, we fuse current information with the previous.

$$\mathbf{F}'_i = \alpha \mathbf{F}_i + (1 - \alpha)\mathbf{F}'_{cur}$$
$$\mathbf{W}'_i = \alpha \mathbf{W}_i + (1 - \alpha)\mathbf{W}'_{cur} \quad (8)$$

Where $\alpha$ is a hyperparameter between [0, 1] regarding how much previous information needs to be saved. Finally, $\mathbf{F}'_i$ and $\mathbf{W}'_i$ will be applied to eq. 4 as $\mathbf{W}$ and $\mathbf{F}$. Note that $\mathbf{F}'_i$ is used to gain the optimal $\mathbf{W}'_i$, but **only** the part of $\mathbf{W}_{cur}$ will be updated and used to weight sample loss.

To enhance the applicability of COP, we use a hyperparameter called COP rate to adjust the degree of influence of COP during training. After acquiring the optimal weights, the loss function is as follows:

$$\hat{w_i}' = (1 - r_{COP}) + r_{COP} * \hat{w_i}$$
$$Loss_{COP} = \sum_{i=0}^{n} \hat{w_i}' Loss_{orig}(p_i, t_i) \quad (9)$$

Where $\hat{w_i} \in \widehat{\mathbf{W}}$ is the final weight of the *i-th* sample; $r_{COP} \in (0, 1]$ indicates COP rate.

### 3.3 Variance Optimizing

Inspired by causal intervention (Pearl et al., 2016), we construct a causal graph (Fig. 2) regarding the causality from feature vectors to the final targets. To clarify our objective, in our causal analysis, we focus on the mitigation of bias arising from the long-tail distribution. This bias tends to make labels with a substantial number of samples more readily predictable during the inference process.

Moreover, the effectiveness of mitigating this bias can be easily quantified.

As shown in Fig. 2, $F \leftarrow L \rightarrow P$ is a backdoor path, a path that connects $F$ and $P$ but does not start with $F$, from $F$ to $P$ in which $F$, $L$, and $P$ are feature vectors, real labels, and model predictions respectively. Thus, the label distribution from the datasets is the confounder between the feature vectors and the final predictions, creating a spurious correlation that disturbs model inference and causes lower performance. For a better understanding of the causal graph, we introduce it as follows: The labels in datasets are the optimizing target of model training ($L \rightarrow P$) which also takes effect on the feature vector learning ($L \rightarrow F$). Appendix. A shows more detail of the causal graph.

In order to remove the confounder caused by backdoor path $F \leftarrow L \rightarrow P$ and make the model learn direct causality between feature vectors and predictions, we introduce $do$-operator (Pearl et al., 2016) on feature vectors $F$ to gain the causality of $F$ on $P$ called $p(P|do(F))$. As shown in Fig. 4 right, through $do$-operator, the edge causing confounder, $F \leftarrow L \rightarrow P$, is blocked, so that the spurious correlation has been removed.

Backdoor adjustment (Pearl, 1995) is a method to reduce confounders in causal inference, and it can perform $do$-operator on the observational datasets. Formally, the backdoor adjustment reduces the bias as:

$$p(P|do(F)) = \sum_{L} p(P|F, L)p(L) \quad (10)$$

Where the detail for the equation is in Appendix B.

Inspired by the eq. 10, it can reduce the spurious correlation if we process the feature vectors of the same label separately. So we propose a method to optimize the feature vectors in the same label by using the property of variance.

When the fluctuation among samples is serious, the variance will be large, and vice versa. In other words, we can minimize the variance between the samples of the same label to bring them closer together and at the same time achieve intervention. Along this way, we introduce a new loss function using the variance to optimize the feature representation as follows:

$$Loss_{VOP} = \sum_{l=0}^{L} \sum_{n=0}^{N} var(\mathbf{F}_{(l,n)}) \quad (11)$$

Where $L$ is the label amount in one batch of data, $N$ is the dimension amount of feature vectors, and

$\mathbf{F}_{(l,n)}$ means the n-th dimensional feature of all samples of label $l$. $var(\cdot)$ is the operation of computing sample variance. Note that missing several labels is possible in one batch of data.

This method is called variance optimizing (VOP). Additionally, benefiting from the random mechanism of mini-batch gradient descent, one sample can combine with different samples into the same batch in each training epoch, which brings two main advantages as follows: Firstly, the diversity of sample combination is ensured automatically. Secondly, the impact of abnormal samples is mitigated due to the overall optimization.

### 3.4 Overall Optimization Framework

We combine COP and VOP into a unified framework called covariance and variance optimization framework (OVO), as shown in Fig. 3, and the algorithm. 1 (Appendix. C) illustrates the detailed training procedure of OVO. The final loss is as follows:

$$Loss_{final} = Loss_{COP} + \lambda Loss_{VOP} \quad (12)$$

Where $\lambda$ is a hyperparameter for the weight of $Loss_{VOP}$.

## 4 Experiment

### 4.1 Dataset

We conduct experiments to evaluate our approach on two popular joint entity and relation extraction datasets: ACE05[1] and SciERC[2](Luan et al., 2018). For ACE2005, we use the same split following (Luan et al., 2019) in the train, development, and test sets. ACE2005 dataset includes various domain data, such as newswire and broadcast news, while the SciERC dataset includes abstracts collected from AI conferences/workshops.

Additionally, to evaluate the effectiveness of OVO in out-of-distribution settings, we conduct experiments on CrossRE [3] (Bassignana and Plank, 2022) which is a dataset for the relation extraction task across 6 different domains, namely news, politics, natural science, music, literature, and artificial intelligence.

### 4.2 Evalution metrics

We follow the evaluation metrics in previous close works to evaluate the experiment results' performance using the micro F1 measure. For the NER

task, we use the span-level benchmark to evaluate, which one entity is considered recognized correctly if and only if both its boundary and type are precisely predicted. For the RE, we adopt two evaluation metrics as follows: (1) *boundaries* **evaluation (Rel)**: a relation is considered recognized correctly if and only if the boundaries of head and tail entities are correct, and their relation label is correct; (2) *strict* **evaluation (Rel+)**: in addition to the requirement of Rel, the types of two entities are also needed to be correctly predicted.

### 4.3 Implementation Details

We apply OVO to 3 of out of top-5 strong baselines[4], specifically PURE (Zhong and Chen, 2021), SpERT.PL (Sai et al., 2021), and PL-Marker (Ye et al., 2022). For ACE2005, the BERT encoders are *bert-base-uncased* (Devlin et al., 2019) and *albert-xxlarge-v1*(Lan et al., 2020) which are in the common domain. For SciERC, the BERT encoder is *scibert-scivocab-uncased* (Beltagy et al., 2019) which is in the scientific domain. We use our proposed method, OVO, to optimize the feature vectors as the extra plugin beyond the common model training with all hyperparameters the same as the base models. To ensure robustness, we conduct experiments with 3 different seeds and report the average scores.

### 4.4 Baselines

For comparison, we choose serials of state-of-the-art models as follows: (1) **PURE** (Zhong and Chen, 2021) uses two independent models for NER and RE respectively in a frustratingly easy way, and is the first to propose levitated markers for RE task called PURE(Approx) to accelerate the inference speed, but with a slight drop in performance; (2) **SpERT.PL** (Sai et al., 2021) uses part-of-speech and logits of entities to enhance the features of entities and relations which shows strong performance in the SciERC dataset. (3) **PL-Marker** (Ye et al., 2022) is the first to leverage levitated markers to both NER and RE tasks using a packed method with great improvement.

Additionally, for comparison with other causal debiasing methods, we choose two related methods as follows: (1) **CFIE** (Nan et al., 2021) uses causal inference to tackle the long-tailed IE issues; (2) **CoRE** (Wang et al., 2022) designs a counterfactual

---

[1] https://catalog.ldc.upenn.edu/LDC2006T06
[2] http://nlp.cs.washington.edu/sciIE/
[3] https://github.com/mainlp/CrossRE

[4] https://paperswithcode.com/sota/joint-entity-and-relation-extraction-on

analysis to debias RE by reducing the side-effects of entity mentions.

### 4.5 Results

Table 1 shows the results of comparative experiments between our proposed method and baselines. As is shown, with the *scibert-scivocab-uncased* encoder, our proposed method, OVO, outperforms PURE-F, SpERT.PL, and PL-Marker by +3.3%, +2.1%, and +2.9% with **Rel+** on SciERC respectively. Furthermore, on ACE2005, with both *bert-base-uncased* encoder and *albert-xxlarge-v1* encoder, our method also outperforms the baselines on **Rel+**. Such improvements over baselines indicate the effectiveness and generalization of OVO.

Additionally, due to CoRE (Wang et al., 2022) being only for RE, we compare OVO with CFIE (Nan et al., 2021) and CoRE only in RE using the result of OVO in NER. For the base model, we chose PURE because of SpERT.PL is a joint model that cannot handle NER and RE tasks separately and PL-Marker uses a packed method that cannot handle only one pair of entities in RE. As shown in Table 1, CFIE and CoRE can improve performance somewhat but are still lower than OVO. The results can show the effectiveness of OVO compared with other methods. Specifically, on SciERC, the REL+ of OVO was 1.6% higher than CFIE; on Ace2005, the REL+ of OVO was 0.2% higher than CoRE.

To evaluate the effectiveness of OVO on out-of-distribution, we conduct experiments on an out-domain setting as follows: using the train set of one domain during training, and using the test sets of the other five domains during testing. We choose PL-Marker as the base model and then apply OVO on each domain. As shown in Table. 2, using OVO gains improvements in all cases with an average improvement of 1.5% and a best improvement of 2.4% in the ai domain.

### 4.6 Ablation Study

In this section, we conduct ablation studies to elucidate the effectiveness of different components of our proposed method on the baseline PL-Marker (Ye et al., 2022). We use the evaluation metrics the same as in Table 1 on both ACE2005 and SciERC datasets using *bert-base-uncased* and *scibert-scivocab-uncased* as bert encoders respectively.

**w/o. COP:** To evaluate the effectiveness of covariance optimizing, we remove COP and only use VOP in the model training process and set all weights of feature vectors as 1 for computing $Loss_{COP}$. As shown in Table 3, the performance of **Rel** and **Rel+** drop 0.8%-2.2% without COP, which can represent the importance to reduce the spurious correlation among features.

**w/o. VOP:** To verify the effectiveness of variance optimization, we make VOP disabled and let only COP work in the training procedure. As shown in Table 3, when removing VOP, the performance of **Rel** and **Rel+** drop 2.0%-3.4% and 1.6%-2.2% respectively, which can indicate the effect of the causal intervention and the VOP play an important role in OVO.

**w. SNet:** StableNet (Zhang et al., 2021) is similar work with COP which use Random Fourier Features to enhance the non-linearity of enhanced features. To compare the effectiveness between COP and StableNet, we replace COP with StableNet keeping other settings the same, and then negotiate the final performance between them. The performance shows that using COP gains +0.7% and +0.4% improvement in **Rel+** on ACE2005 and SciERC respectively. Moreover, from the results in Table 5, the consumption of COP is much smaller than that of StableNet.

## 5 Analysis

### 5.1 Fine-grained analysis

To analyze the debias ability of OVO, we conduct experiments to evaluate the performance of each label on PL-Marker. The detailed results for each relation label on SciERC are shown in Table 4. From the table, the number of **USERD-FOR** accounts for more than half, 54.7% of the total, but that of **COMPARE** is only 3.9% of the total.

From the results, the two worst-performing labels both have a small number of samples which are only 6.9% and 6.4% of the total, respectively. For the comparison between with or without OVO, we find that the performances in all labels gain improvement, except for **USED-FOR**, which is one that has the most samples in the dataset and keeps comparable performance. Additionally, the improvement increases as the number of samples decreases. The most improvement is on **PART-OF**, which outperforms +9.1% in **F1** than without VOP. Moreover, the overall results of average **F1** gain +4.0% improvement.

| | Encoder | ACE2005 | | | SciERC | | |
|---|---|---|---|---|---|---|---|
| | | **Ent** | **Rel** | **Rel+** | **Ent** | **Rel** | **Rel+** |
| SPtree (Miwa and Bansal, 2016) | $LSTM$ | 83.4 | - | 55.6 | - | - | - |
| DyGIE (Luan et al., 2019) | $ELMo$ | 88.4 | 63.2 | - | 65.2 | 41.6 | - |
| OneIE† (Lin et al., 2020) | $B_L$ | 88.8 | 67.5 | - | - | - | - |
| DyGIE++† (Wadden et al., 2019) | $B_B/SciB$ | 88.6 | 63.4 | - | 67.0* | 47.5* | - |
| SPERT (Eberts and Ulges, 2020) | | 86.8* | 63.8* | - | 70.3 | 50.8 | - |
| UniRE† (Wang et al., 2021) | | 88.8 | 65.7* | 64.3 | 68.4 | 40.2* | 36.9 |
| SpERT.PL (Sai et al., 2021) | | 87.4* | 65.3* | 62.1* | 70.5 | 51.3 | 39.4 |
| +OVO | | **87.9** | **67.9** | **65.3** | **71.7** | **52.6** | **41.5** |
| PL-Marker† (Ye et al., 2022) | | **89.8** | 69.0 | 66.5 | 69.9 | 53.2 | 41.6 |
| +OVO† | | 89.6 | **70.6** | **67.7** | 70.7 | **56.1** | **44.5** |
| PURE-F† (Zhong and Chen, 2021) | $B_B/SciB$ | 90.1 | 67.7 | 64.8 | 68.9 | 50.1 | 36.8 |
| +CFIE† ($Nan\,et\,al.$, 2021) | | - | 67.7 | 64.5 | - | 50.9 | 38.5 |
| +CoRE† ($Wang\,et\,al.$, 2022) | | - | 67.6 | 64.9 | - | 50.7 | 38.0 |
| +OVO† | | **90.5** | **68.0** | **65.1** | **70.0** | **51.6** | **40.1** |
| UniRE† (Wang et al., 2021) | $ALB_{XX}$ | 90.2 | - | 66.0 | - | - | - |
| PURE-F† (Zhong and Chen, 2021) | | 90.9 | 69.4 | 67.0 | - | - | - |
| PL-Marker† (Ye et al., 2022) | | 91.1 | 73.0 | 71.1 | - | - | - |
| +OVO† | | **91.2** | **74.6** | **72.3** | - | - | - |

Table 1: Overall test results of F1 in joint entity and relation task on the dataset of ACE2005 and SciERC. †: the model or method leverages cross-sentence information as extra features. The encoders used in different models: $B_B$ = $bert\text{-}base\text{-}uncased$, $B_L$ = $bert\text{-}large\text{-}uncased$, $SciB$ = $scibert\text{-}scivocab\text{-}uncased$, $ALB_{XX}$ = $albert\text{-}xxlarge\text{-}v1$. Model name abbreviation: PURE-F = PURE(Full). Evaluation metric abbreviation: **Rel** denotes the boundaries evaluation and **Rel+** denotes the strict evaluation. *: the result is not given in the original works, and we run the corresponding experiment, reporting the results in the table.

| Rel(%) | news | science | politics | literature | ai | music | avg |
|---|---|---|---|---|---|---|---|
| PL-Marker | 9.7 | 13.0 | 14.4 | 14.8 | 11.6 | 14.1 | 12.9 |
| +OVO | **10.3** | **14.3** | **15.4** | **16.4** | **14.0** | **16.4** | **14.5** |

Table 2: Results in the out-domain setting on the CrossRE dataset.

| | ACE2005 | | | SciERC | | |
|---|---|---|---|---|---|---|
| | **Ent** | **Rel** | **Rel+** | **Ent** | **Rel** | **Rel+** |
| **OVO** | 89.6 | **70.6** | **67.7** | **70.7** | **56.1** | **44.5** |
| **w/o. COP** | **89.7** | 69.7 | 66.9 | 70.0 | 54.1 | 43.3 |
| **w/o. VOP** | 89.5 | 68.6 | 65.5 | **70.7** | 52.7 | 42.9 |
| **w. SNet** | 89.4 | 70.1 | 67.0 | 70.5 | 55.0 | 44.1 |

Table 3: Ablation on ACE2005 and SciERC test set with different component setups. w. SNet means replacing COP with StableNet in model training.

## 5.2 Effect of Covariance Optimizing

As the results are shown in Table 5, by using StableNet, it gains +0.6% improvement on **Ent**, and COP improves on it +0.2%. Table 3 shows more results between COP and StableNet. So, we can conclude that stable learning technology can actually achieve improvement for our task. But the consumption of computing resources is relatively high and it costs approximately 5 and 1.7 times for training time and GPU memory respectively than the original. In comparison, COP uses fewer resources but gets better performance. Additionally, more detailed results of hyperparameters are

reported in Appendix. D.

## 5.3 Effect of Variance Optimizing

To enhance adaptation to diverse methods, we introduce a weight for $Loss_{VOP}$ to mitigate its impact on the normal training of the model and avoid performance degradation. The optimal weight varies depending on the specific dataset and method employed, as different weights yield the best performance. The detailed results are reported in Appendix. E.

## 6 Conclusion

In this work, we propose the OVO, a novel model training framework for optimizing feature vectors using causal technology. There are two main parts of OVO which are called COP and VOP. From the perspective of the correlation among features, COP reduces the spurious relation by minimizing the covariance computed from all features. From the viewpoint of each feature, VOP makes the feature more causal by minimizing the variance calculated from each feature of the same label. After optimizing the features toward causal, our method achieves

| | Metric (Rel) | UF | CON | EF | HO | PO | FO | COM | avg. |
|---|---|---|---|---|---|---|---|---|---|
| **Proportion (%)** | - | 54.7 | 12.6 | 9.3 | 6.9 | 6.4 | 6.0 | 3.9 | - |
| **w/o. OVO (%)** | Recall | 55.2 | 62.6 | 46.2 | 64.2 | 28.6 | 23.7 | 52.6 | 47.6 |
| | Precision | 53.6 | 55.0 | 51.2 | 58.1 | 48.7 | 25.9 | 50.0 | 48.9 |
| | F1 | **54.3** | 58.6 | 48.6 | 60.1 | 36.0 | 24.8 | 51.3 | 47.7 |
| **w. OVO (%)** | Recall | 53.1 | 65.7 | 49.5 | 70.2 | 36.5 | 25.4 | 55.3 | 50.8 |
| | Precision | 55.4 | 53.3 | 54.2 | 59.5 | 59.0 | 35.7 | 60.0 | 53.9 |
| | F1 | 54.2 | **58.9** | **51.7** | **64.4** | **45.1** | **29.7** | **57.5** | **51.7** |
| | Δ | -0.1 | +0.3 | +3.1 | +4.3 | +9.1 | +4.9 | +6.2 | +4.0 |

Table 4: The fine-grained results on the SciERC dataset. Relation label abbreviation: UF=USED-FOR, CON=CONJUNCTION, EF=EVALUATE-FOR, HO=HYPONYM-OF, PO=PART-OF, FO=FEATURE-OF, COM=COMPARE. Proportion means the percentage of the number of one relation label in total data.

| Name | Ent | GPU (GB) | Time (h) |
|---|---|---|---|
| **PL-Marker** | 69.9 | *19* | *2* |
| **w. COP** | *70.7* | 24 | 4 |
| **w. SNet** | 70.5 | 32 | 10 |

Table 5: The experiment results on the SciERC. The base model is PL-Marker. COP means covariance optimizing; Time (h) is the time consumption of model training. The text with *bold and italics* indicates the best performance, and that with underlining indicates the second best.

improvements on 3 strong baselines and reduces the influence of data unbalanced bias. In the future, we will pay more attention to (1) how to further reduce resource consumption on COP and (2) how can let the weight of VOP adjusts itself automatically to adapt to the different datasets.

## Limitations

OVO is a novel training framework for optimizing feature representations directly using causal technology. The limitations of OVO are as follows:

- COP is an additional training process in each batch that will improve resource consumption. Larger models and more previous information used in COP will cost more additional resources.

- To adapt OVO to more models, there are some hyperparameters to control the degree of effect of COP and VOP during training. These hyperparameters are sensitive to different models and datasets, so it takes time to experiment to find the optimal values of them.

## Acknowledgements

The State Key Program of National Natural Science of China, Grant/Award Number:61533018; National Natural Science Foundation of China, Grant/Award Number: 61402220; The Philosophy and Social Science Foundation of Hunan Province, Grant/Award Number: 16YBA323; Natural Science Foundation of Hunan Province, Grant/Award Number: 2020JJ4525,2022JJ30495; Scientific Research Fund of Hunan Provincial Education Department, Grant/Award Number: 18B279,19A439,22A0316.

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

# A  Causal Graph

From Fig. 2, $L \rightarrow F$ denotes that real labels in datasets have an effect on the feature vectors learned by model training, and $L \rightarrow P$ denotes that real labels have an effect on model predictions because the loss in training is calculated by labels and predictions. So the backdoor path $F \leftarrow L \rightarrow P$ makes models give relatively high scores to the labels with numerous samples.

## B   Derivation Process of Backdoor Adjustment

In this section, we will give an explanation for each step of backdoor adjustment. To make it easier to understand, we draw a new equation as follows:

$$
\begin{aligned}
p(P|do(F)) &\overset{(1)}{=} \sum_L p(P|do(F),\underline{L})p(\underline{L}|do(F)) \\
&\overset{(2)}{=} \sum_L p(P|do(F),L)P(L|\underline{F}) \\
&\overset{(3)}{=} \sum_L p(P|\underline{F},L)p(L|F) \\
&\overset{(4)}{=} \sum_L p(P|F,L)\underline{p(L)}
\end{aligned}
\tag{13}
$$

Where the meaning of $P$, $F$, and $L$ are the same as Fig. 2; $do(\cdot)$ means $do$-operator. The underline means the main differences from the previous step.

(1) using the total probability theorem where $do(F)$ can be seen as a normal random variables event like $P$ and $L$.

(2) from the causal graph in Fig. 2, the path $L \rightarrow F$ is a direct causal path without backdoor paths, so $p(L|do(F))$ and $p(L|F)$ are equivalent.

(3) because of conditioning on $L$, the association in the path $F \leftarrow L \rightarrow P$ is blocked. So in this case, the association in the path $F \rightarrow P$ is causality which can remove the $do$-operator without any influence.

(4) due to the direction of the causal path between $L$ and $F$ being $F \leftarrow L$, there is no association from $F$ to $L$ which causes the $L$ to be independent with $F$ meaning $p(L|F) = p(L)$.

## C   Overall Algorithm of OVO

The algorithm of OVO is shown in the algorithm. 1.

## D   Analysis of COP

### D.1   Saving Batch Number

When the dataset is large enough, it is hard to put all feature vectors into COP and optimize them at once. So, by using eq. 6, we can use part of the vectors to relieve the pressure on computing resources. To research the effect of COP when just using partial vectors, we conduct comparative experiments with

---

**Algorithm 1** The training procedure of the OVO

**Input:** Initial pre-trained BERT encoder $\mathbf{BE}^{(0)}$ and classifier $\mathbf{C}^{(0)}$, training dataset $\mathcal{D}$, training epochs for model training $T_{mt}$, training epochs for COP $T_{cop}$
**Output:** Trained model $(\mathbf{BE}^{(T_{mt})}, \mathbf{C}^{(T_{mt})})$
1: **for** $t_{mt} = 0$ **to** $T_{mt}$ **do**
2:      $\mathbf{F_{pre}} \leftarrow$ Clear the previous feature vectors;
3:      $\mathbf{W_{pre}} \leftarrow$ Clear the previous weights;
4:      **for** $(Sents, Spans)$ in $\mathcal{D}$ **do**
5:          $\mathbf{H} = \mathbf{BE}^{(t_{mt})}(Sents)$
6:          $\mathbf{F_{cur}} = GetSpanFeatures(\mathbf{H})$
7:          $\mathbf{P} = \mathbf{C}^{(t_{mt})}(\mathbf{F_{cur}})$
8:          $\mathbf{W_{cur}} \leftarrow$ Initial
9:          $\mathbf{F} = concat(\mathbf{F_{pre}}, \mathbf{F_{cur}})$
10:         $\mathbf{W} = concat(\mathbf{W_{pre}}, \mathbf{W_{cur}})$
11:         **for** $t_{cop} = 0$ **to** $T_{cop}$ **do**
12:            $\mathcal{L_W} = \sum_{i \neq j} cov(\mathbf{WF}_i, \mathbf{WF}_j)$
13:            $\mathbf{W_{cur}} \leftarrow Optim(\mathcal{L_W})$
14:         **end for**
15:         $(\mathbf{F_{pre}}, \mathbf{W_{pre}}) \leftarrow$ update and fuse
16:         $\mathcal{L}_{orig} = CrossEntropy(\mathbf{P}, Spans)$
17:         $\mathcal{L}_{COP} = \mathbf{W_{cur}}\mathcal{L}_{orig}$
18:         $\mathcal{L}_{VOP} = \sum_{l=0}^{L}\sum_{n=0}^{N} var(\mathbf{F}_{(l,n)})$
19:         $\mathcal{L} = \mathcal{L}_{COP} + \mathcal{L}_{VOP}$
20:         $(\mathbf{BE^{t_{mt}+1}}, \mathbf{C^{t_{mt}+1}}) \leftarrow Optim(\mathcal{L})$
21:      **end for**
22: **end for**

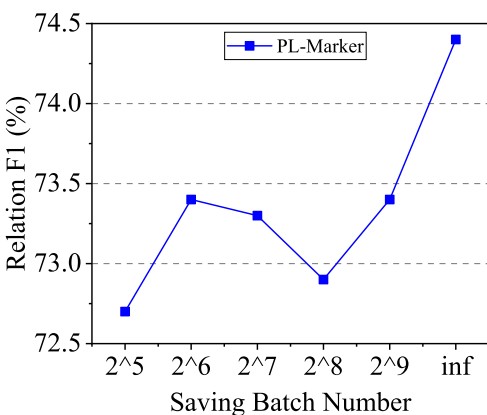

Figure 5: The impact of different Saving Batch Numbers on the performance of COP is evaluated using the SciERC test set. The *inf* notation denotes the scenario where all feature vectors are used for COP without any deletions. Relation F1 means evaluating with gold entities in **Rel+**; for eq. 8, $\alpha$ is equal to 0.9 in the experiments.

different saving batch numbers on PL-Marker (Ye et al., 2022). As shown in Fig. 5, as the saving batch number rises, the overall trend of F1 also rises and reaches the best performance at $inf$ (around $2^{10}$ in our training setting). It can be concluded that using more vectors for COP can reduce spurious correlation better and gain more improvement, but require more computing resources.

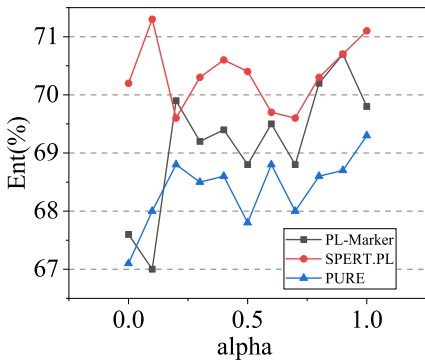

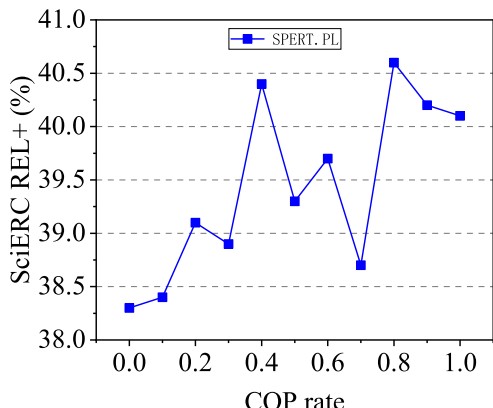

Figure 7: Effect of the different weights of VOP. F1 is Rel on datasets.

0.4 on ACE2005 and SciERC respectively. The detailed results for the COP rate are shown in Fig. 7 which only uses COP in experiments.

# E Analysis of VOP

## E.1 Tradeoff on Variance Optimizing

For better adaptation to different tasks, it is important to give $Loss_{VOP}$ a weight, preventing it to influence the model training and hurt the performance. We compute the final loss with weighed $Loss_{VOP}$ as follows:

$$Loss_{final} = Loss_{COP} + \lambda Loss_{VOP} \quad (14)$$

Where $\lambda$ is a hyperparameter regarding the weight of $Loss_{VOP}$. From the results in Fig. 8, different datasets get the best performance in different weights of VOP which $\lambda = 0.2$ is the optimal value for ACE2005 and $\lambda = 0.1$ is that for SciERC, and the performance is particularly sensitive to the weight which changes drastically using different $\lambda$.

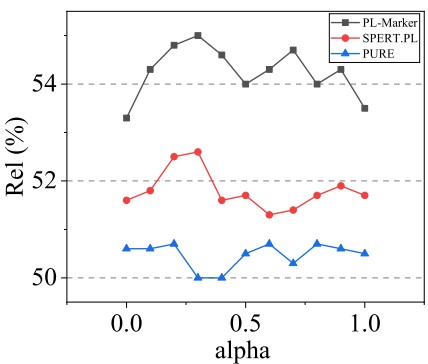

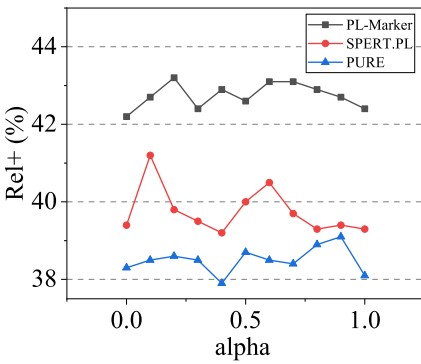

Figure 6: Effect of different $\alpha$ of COP.

## D.2 Effect of alpha

To explore the effect of fusing current information, we conduct experiments for fusing hyperparameter $\alpha$ of eq. 8. We fix the saving batch number to infinite and change the $\alpha$ in the same training configuration. Fig. 6 shows the detailed result of the experiments.

## D.3 COP rate

We use $r_{COP} = 1$ in PURE (Zhong and Chen, 2021) and PL-Marker (Ye et al., 2022). For SpERT.PL, we set the COP rate equal to 0.1 and

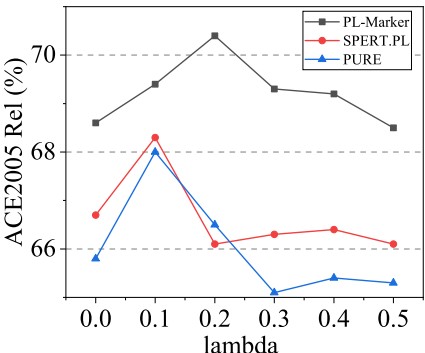

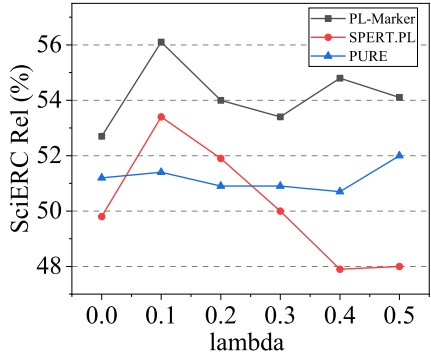

Figure 8: Effect of the different weights of VOP. F1 is Rel on datasets.