# OpenReview forum: "CoVariance-based Causal Debiasing for Entity and Relation Extraction"
_EMNLP/2023/Conference — EMNLP 2023 Findings_

### Official Review · Reviewer_5RrX · 2023-08-03

**Typos Grammar Style And Presentation Improvements:** 1. Since W is a vector in Eq. 4, bett…
**Soundness:** 3

**Excitement:**

4: Strong: This paper deepens the understanding of some phenomenon or lowers the barriers to an existing research direction.

**Paper Topic And Main Contributions:**

Entity and relation extraction models often suffer from a variety of dataset biases, leading to reduced robustness and generalization. To tackle this problem, this paper presents a novel causal framework featuring two techniques, covariance optimization and variance optimization, both aiming to mitigate spurious correlations. Experiments on two benchmarks show that the proposed framework show improved results when integrated into various backbone models.

**Questions For The Authors:**

1. In the causal graph, why B leads to L? Does it mean the real label is only based on bias? Also, does it mean there is no bias in features?
2. In the equation in line 245, are there constraints on W? At least they should be nonnegative and sum up to 1?
3. How is the solution of W calculated?
4. In line 297, adding two features or weights of different samples does not make sense.

**Reasons To Accept:**

1. Mitigating spurious correlations in information extraction models is an important problem, especially for real-world applications where the data is often out-of-distribution. However, this problem is not sufficiently addressed in the literature, and this paper makes valuable contributions to this topic.
2. Previous methods for mitigating spurious correlations for IE models are mainly conducted in the inference phase, this paper introduces innovative techniques to tackle this problem during the training phase.

**Reasons To Reject:**

1. The main claim of reducing dataset bias is not sufficiently supported in experiments. Current experiments only include results on in-distribution test data, long-tailed classes, and some analysis of computational efficiency. However, I think a more important experiment to show the results on out-of-distribution data, which can be another domain of text (with the same entity/relation types), unseen entities, noisy labels, etc.
2. The motivation behind the variance optimization method needs to be clarified. I do not understand why reducing variance is related to backdoor adjustment.
3. Properly defining and consistently using symbols is crucial. Some symbols in the paper are not defined, and some are used more than one time.

Edit after rebuttal: As the authors have added results on OOD examples, I raised the soundness score.

**Reproducibility:**

3: Could reproduce the results with some difficulty. The settings of parameters are underspecified or subjectively determined; the training/evaluation data are not widely available.

**Reviewer Confidence:**

4: Quite sure. I tried to check the important points carefully. It's unlikely, though conceivable, that I missed something that should affect my ratings.

---

> ### Author Rebuttal · Authors · 2023-08-29
>
> Thank you for taking the time to review our paper. We appreciate your insightful comments and valuable feedback, which have undoubtedly improved the quality of our work. In response to your review, we would like to address each of your points in detail.
>
> 1. In the context of out-of-distribution settings, we extend our apologies for the omission of experiments. This omission stems from the focus of our research on the conventional task of joint entity and relation extraction. To supplement the corresponding experiments, we conduct experiments in dataset CrossRE [1] which is a cross-domain dataset for relation extraction. There are six domains in CrossRE, namely news, politics, science, music, literature, and ai.
>
>    Firstly, we apply OVO to the baseline in [1] to evaluate the effectiveness of our proposed method. The task of baseline is in-domain Relation Classification which needs to predict the relation type of entity pairs having a relation. As shown in (Supplement Table D1), OVO gains improvements in all domains of CrossRE with an average improvement of 0.8% and a best improvement of 1.6% in the news domain.
>
> (Supplement Table D1): Results in CrossRE baseline. $w.$ and $w/o.$ mean using OVO or not.
>
> | MICRO-F1(%) | news | politics | science | music | literature | ai   | avg  |
> | ----------- | ---- | -------- | ------- | ----- | ---------- | ---- | ---- |
> | w.o.        | 46.4 | 58.3     | 40.1    | 76.0  | 67.7       | 45.4 | 55.7 |
> | w.          | 48.0 | 58.8     | 40.5    | 77.1  | 68.5       | 46.2 | 56.5 |
>
>
> ​    Then, due to the missing Relation Extraction results in [1], we conduct experiments in CrossRE dataset using PL-Marker [2] in two out-domain settings:
>
> - 1) Overall out-domain setting: using the train set of one domain during training, using the test sets of the other five domains during testing.
>
> - 2) Fine-grained out-domain setting: using the train set of one domain during training, using the test set of another different domain during testing.
>
>   In the overall out-domain setting, as shown in (Supplement Table D2), using OVO gains improvements in all cases ith an average improvement of 1.5% and a best improvement of 2.4% in the ai domain.
>
> (Supplement Table D2): Results in overall out-domain setting. $w.$ and $w/o.$ mean using OVO or not.
>
> | Rel(%) | news | science | politics | literature | ai   | music | avg  |
> | ------ | ---- | ------- | -------- | ---------- | ---- | ----- | ---- |
> | w/o.   | 9.7  | 13.0    | 14.4     | 14.8       | 11.6 | 14.1  | 12.9 |
> | w.     | 10.3 | 14.3    | 15.4     | 16.4       | 14.0 | 16.4  | 14.5 |
>
> In the fine-grained out-domain setting, as shown in (Supplement Table D3), due to the limitation of time, we only conduct experiments in the news and science domains. The results show the out-domain effectiveness of OVO.
>
> (Supplement Table D3): Results in fine-grained out-domain setting. All values in the table are Rel(%). $w.$ and $w/o.$ mean using OVO or not; | means the same as above.
>
> | train | test       | w.o  | w.   | train   | test       | w.o  | w.   |
> | ----- | ---------- | ---- | ---- | ------- | ---------- | ---- | ---- |
> | news  | ai         | 10.1 | 10.3 | science | news       | 22.9 | 24.6 |
> | \|    | literature | 7.5  | 8.8  | \|      | ai         | 15.6 | 22.3 |
> | \|    | music      | 6.2  | 7.1  | \|      | literature | 10.2 | 11.3 |
> | \|    | science    | 9.6  | 11.8 | \|      | music      | 9.7  | 11.4 |
> | \|    | polotics   | 14.2 | 16.7 | \|      | polotics   | 16.6 | 18.2 |
>
> The results in (Supplement Table D2) and (Supplement Table D3) show the out-of-distribution effectiveness of OVO.
>
> 2. In datasets with a long-tail distribution of labels, where certain labels are more common and others are rarer, machine learning models tend to favor predicting the more abundant labels to optimize accuracy. To capture this phenomenon in our causal graph, we introduce an edge labeled L -> F to represent how the label distribution influences the feature vectors of data samples.
>
>    Our VOP approach addresses this issue by minimizing variance among samples of the same label, effectively reducing the impact of the long-tail label distribution on feature vectors. As a result, the effect denoted as L -> F is blocked, approximating a do-operator. The outcomes of this approach, as demonstrated in Table 4 of our paper, show significant performance improvements for labels that are less common in the dataset.
>
> 3. We appreciate your feedback regarding the need for proper symbol definitions and consistent usage throughout the paper. We will thoroughly review the paper to identify any undefined symbols and ensure that all symbols are used consistently and appropriately.
>
> **[Questions]**
>
> 1. In the causal graph, why B leads to L? Does it mean the real label is only based on bias? Also, does it mean there is no bias in features?
>
>    B -> L means the bias affects the models through labels. The bias of long-tail distribution is caused by data collection, which produces large differences in sample sizes under different labels. This phenomenon will influence the model during training, which leads to labels with a greater abundance of samples tending to be predicted more often by the model.
>
>    It does not mean the real label is only based on bias, but bias in the data collection process contributes to the long-tail distribution of labels. The features are initially without bias due to the random initialization, but the bias will further affect features during training.
>
> 2. In the equation in line 245, are there constraints on W? At least they should be nonnegative and sum up to 1?
>
>    We use the softmax function to constrain the values of weights, which seem missing in the paper. We apologize for this and will fix it in the paper.
>
> 3. How is the solution of W calculated?
>
>    We use an extra training phase to acquire the final weight using gradient descent. Appendix C provides an overall algorithm of OVO and lines 11-14 show how to acquire the final weights.
>
> 4. In line 297, adding two features or weights of different samples does not make sense.
>
>    Due to the continuous updating of features during training, feature inconsistency may arise. Line 297 combines the features from the current batch with those saved from previous batches to mitigate the impact of previous features and ensure feature consistency. Therefore, in this context, adding the features or weights of different samples is meaningful as it contributes to maintaining feature consistency and improving the overall performance of the model. Additionally, Appendix D.2 illustrates the effect of the fusion operation.
>
> **[references]**
>
> [1] Elisa Bassignana, Barbara Plank. CrossRE: A Cross-Domain Dataset for Relation Extraction[J], EMNLP (Findings), 2022: 3592-3604.
>
> [2] Deming Y, Yankai L, Peng L, Maosong S, et al. Packed Levitated Marker for Entity and Relation Extraction[C], Annual Meeting of the Association for Computational Linguistics, 2022, Proceedings of the 60th Annual Meeting of the Association for Computational Linguistics (Volume 1: Long Papers): 4904–4917.

---

### Official Review · Reviewer_qNHL · 2023-08-04

**Soundness:** 4

**Excitement:**

4: Strong: This paper deepens the understanding of some phenomenon or lowers the barriers to an existing research direction.

**Paper Topic And Main Contributions:**

This paper introduces a covariance and variance optimization framework (OVO) to optimize feature representations from a causality perspective for joint entity and relation extraction tasks. Inspired by stable learning, they propose covariance optimizing (COP) to reduce collinearity of each feature, get the causal effects, alleviate the data selection and distribution bias. In addition, they propose variance optimizing (VOP) to minimize the variance of each dimension in the feature vectors of the same class label, mitigating the long-tail distribution bias further. This framework achieves improvement over 3 strong baselines.

**Reasons To Accept:**

- The proposed framework alleviates data biases and enhances feature representation.
- It achieves noticeable improvement when applying to 3 strong baselines for joint entity and relation extraction task

**Reasons To Reject:**

- The framework involves two optimization problems, which could be slow and expensive. It would be beneficial to include more details such as training time and GPU memory usage.
- It would be better to include analysis for other data biases besides the long-tail distribution bias.

**Reproducibility:**

4: Could mostly reproduce the results, but there may be some variation because of sample variance or minor variations in their interpretation of the protocol or method.

**Reviewer Confidence:**

2: Willing to defend my evaluation, but it is fairly likely that I missed some details, didn't understand some central points, or can't be sure about the novelty of the work.

---

> ### Author Rebuttal · Authors · 2023-08-29
>
> Thank you for taking the time to review our paper. We appreciate your insightful comments and valuable feedback, which have undoubtedly improved the quality of our work. In response to your review, we would like to address each of your points in detail.
>
> **[include more details such as training time and GPU memory usage]**
>
> Thank you for bringing this to our attention. We wholeheartedly agree that providing more comprehensive information about resource consumption is indeed valuable for a better understanding of our proposed method. As illustrated in (Supplement Table C1), we have included a breakdown of resource consumption specifically for the SciERC dataset. We will provide a comprehensive overview of overall resource consumption in the next version of our paper. Your feedback is greatly appreciated, and we are committed to enhancing the quality of our research.
>
> (Supplement Table C1):  Resource consumption of OVO. $w.$ and $w/o.$ mean using OVO or not.
>
> | Name       | Task    | OVO | GPU(GB) | Time(h) |
> |------------|---------|-----|---------|---------|
> | PL-Marker  | NER     | w/o. | 19     | 2.0     |
> |            |         | w.  | 24     | 4.0     |
> |            | RE      | w/o. | 4      | 0.3     |
> |            |         | w.  | 6      | 0.5     |
> | PURE       | NER     | w/o. | 6      | 1.0     |
> |            |         | w.  | 19     | 1.2     |
> |            | RE      | w/o. | 8      | 0.9     |
> |            |         | w.  | 12     | 1.1     |
> | SPERT.PL   | NER+RE  | w/o. | 11     | 0.2     |
> |            |         | w.  | 18     | 0.5     |
>
> **[include analysis for other data biases besides the long-tail distribution bias.]**
>
> We agree with this about analyzing other biases. During rebuttal, we conduct experiments to evaluate the effectiveness of OVO in out-of-distribution.
>
> We conduct experiments in CrossRE[1] dataset using PL-Marker [2] in two out-domain settings:
> - 1) Overall out-domain setting: using the train set of one domain during training, using the test sets of the other five domains during testing.
> - 2) Fine-grained out-domain setting: using the train set of one domain during training, using the test set of another different domain during testing.
>
> In the overall out-domain setting, as shown in (Supplement Table C2), using OVO gains improvements in all cases ith an average improvement of 1.5% and a best improvement of 2.4% in the ai domain.
>
> (Supplement Table C2): Results in overall out-domain setting. $w.$ and $w/o.$ mean using OVO or not.
> |  Rel(%)  | news | science | politics | literature | ai   | music | avg  |
> | ---- | ---- | ------- | -------- | ---------- | ---- | ----- | ---- |
> | w/o. | 9.7  | 13.0    | 14.4     | 14.8       | 11.6 | 14.1  | 12.9 |
> | w.   | 10.3 | 14.3    | 15.4     | 16.4       | 14.0 | 16.4  | 14.5 |
>
> In the fine-grained out-domain setting, as shown in (Supplement Table C3), due to the limitation of time, we only conduct experiments in the news and science domains. The results show the out-domain effectiveness of OVO.
>
> (Supplement Table C3): Results in fine-grained out-domain setting. All values in the table are Rel(%). $w.$ and $w/o.$ mean using OVO or not; | means the same as above.
> | train | test       | w.o  | w.   | train   | test       | w.o  | w.   |
> | ----- | ---------- | ---- | ---- | ------- | ---------- | ---- | ---- |
> | news  | ai         | 10.1 | 10.3 | science | news       | 22.9 | 24.6 |
> | \|    | literature | 7.5  | 8.8  | \|      | ai         | 15.6 | 22.3 |
> | \|    | music      | 6.2  | 7.1  | \|      | literature | 10.2 | 11.3 |
> | \|    | science    | 9.6  | 11.8 | \|      | music      | 9.7  | 11.4 |
> | \|    | polotics   | 14.2 | 16.7 | \|      | polotics   | 16.6 | 18.2 |
>
> The results in (Supplement Table C2) and (Supplement Table C3) show the out-of-distribution effectiveness of OVO.
>
> **[references]**
>
> [1] Elisa Bassignana, Barbara Plank. CrossRE: A Cross-Domain Dataset for Relation Extraction[J], EMNLP (Findings), 2022: 3592-3604.
>
> [2] Deming Y, Yankai L, Peng L, Maosong S, et al. Packed Levitated Marker for Entity and Relation Extraction[C], Annual Meeting of the Association for Computational Linguistics, 2022, Proceedings of the 60th Annual Meeting of the Association for Computational Linguistics (Volume 1: Long Papers): 4904–4917.

---

### Official Review · Reviewer_debR · 2023-08-09

**Soundness:** 2

**Excitement:**

2: Mediocre: This paper makes marginal contributions (vs non-contemporaneous work), so I would rather not see it in the conference.

**Missing References:**




**Paper Topic And Main Contributions:**

This paper analyzes the spurious correlations in entity and relation extraction. The authors propose a covariance optimization method to improve the model performance.

**Reasons To Accept:**

* The results show that covariance optimization helps to improve model performance on entity and relation extraction.

**Reasons To Reject:**

* The causal analysis has flaws. The authors confuse several concepts in the proposed SCM. Is the SCM for the forward or backward pass?
Is each node in the SCM a distribution or a single instance? Is L in the SCM a single label or all the same labels? The authors use these contradicting meanings to explain the SCM in different parts of the paper.
* The authors do not provide convincing evidence to show that the proposed method can be explained by causal theory. To be more specific, equation 11 is not a reasonable realization of equation 10. To perform backdoor adjustment, it is essential to change the value/distribution of the confounder proactively. Minimizing "the variance between the samples of the same label" (line 360) is not a causal intervention based on the proposed SCM.
* The evaluation is not sound. No baseline for bias mitigation is compared. The method is only evaluated in an in-domain setting.

*After several rounds of discussion with the authors, I believe their causal analysis has significant flaws. First, the proposed SCM and follow-up analysis are not coherent. Some of the nodes, links, and terms in the equations do not have concrete meanings, and the authors change the definition of variables in different steps of the proof. For example, the definition of $L$ is inconsistent in the analysis and this results in a weird term $p(P|F,L)$. Second, the most important step that connects the causal analysis and the proposed covariance minimization method is based on a **wild assumption** that a strong correlation between $F$ and $L$ will lead to $p(P|do(F)) = p(P|F) = p(P|F,L)$. This assumption challenges the basic of causal intervention and is unrealistic in nature.*

**Reproducibility:**

3: Could reproduce the results with some difficulty. The settings of parameters are underspecified or subjectively determined; the training/evaluation data are not widely available.

**Reviewer Confidence:**

4: Quite sure. I tried to check the important points carefully. It's unlikely, though conceivable, that I missed something that should affect my ratings.

---

> ### Author Rebuttal · Authors · 2023-08-29
>
> We extend our gratitude for dedicating your time to the examination of our scholarly contribution. In light of your comprehensive review, we hereby express our intention to meticulously address each of the points you have raised, providing a thorough and detailed response.
>
> **[The causal analysis has flaws.]**
>
> We apologize for the slight discussion of the causal analysis. We will detailly describe the causal analysis as follows:
>
> To simplify the description, we use the same abbreviations in the paper:
>
> L : Real Label
> F : Feature vector
> P : Prediction
>
> In our causal graph, P represents the outputs of the model, while L represents the real labels of samples. Throughout the training process, our primary objective is to minimize the disparity between P and L, a task typically quantified using a loss function, such as cross-entropy. Consequently, to capture and depict this relationship between L and P, we introduce an edge L -> P within the causal graph.
>
> Moreover, within datasets characterized by a long-tail distribution of labels, it is observed that labels associated with a higher frequency of occurrence tend to receive more frequent predictions from the model. This observation is a direct outcome of the model's learning process, wherein optimization is primarily directed toward enhancing accuracy with respect to the predominant class. Consequently, to represent the impact of label distribution on the feature vectors of data samples, we introduce an edge, symbolized as L -> F, into our causal graph.
>
> We will answer the questions below this comment as follows:
>
> 1. Is the SCM for the forward or backward pass?
>
>    The proposed SCM is for backward pass. Our initial focus is on the phenomenon that the long-tail distribution within datasets impacts the performance of labels associated with a small number of samples. Subsequently, we undertake a causal analysis to mitigate this influence.
>
> 2. Is each node in the SCM a distribution or a single instance?
>
>    The nodes in the SCM are distributions. Our objective is to analyze the influence of labels on the model training procedure.
>
> 3. Is L in the SCM a single label or all the same labels?
>
>    L represents all labels, encompassing various labels rather than a single label or identical labels. The grouping of samples based on labels is our strategy to alleviate the impact of label distribution, and it is not directly related to SCM.
>
> **[The authors do not provide convincing evidence to show that the proposed method can be explained by causal theory. ]**
>
> To our best knowledge, backdoor adjustments do not necessarily require a change in the value of the data, and statistical methods can be used to control for or account for the effects of confounders by appropriately modeling their influence, without requiring active changes to the value or distribution of the confounders. For example, in Simpson's Paradox, backdoor adjustment is implemented by statistical methods.
>
> Moreover, there are many recent works to introduce causal technology into the deep learning field using different implementation methods. And part of them does not change the value/distribution of the confounder. For example, [1] does not change the dataset, but designs counterfactual decoders for backdoor adjustment; [2] proposes a method to approximate
> the causal intervention for reducing bias influence, and utilize it during training.
>
> VOP utilizes minimizing variance between samples of the same label to equalize the significance of different labels. To do so, the long-tail distribution of labels has reduced the impact on feature vectors. Thus the effect of **L -> F** is blocked which approximates to do-operator. As shown in Table 4 in the paper, the performances of the labels with a fewer number of samples gain significant improvement.
>
> **[The evaluation is not sound. No baseline for bias mitigation is compared. The method is only evaluated in an in-domain setting.]**
>
> For other bias mitigation methods, we have undertaken a comparative analysis between our proposed approach and StableNet which is similar to COP. The outcomes of these experimental comparisons have been documented in Tables 3 and 5. The findings therein affirm that our method outperforms StableNet in terms of both performance metrics and resource consumption.
>
> For out-domain settings, we apologize for missing the corresponding experiments due to the focus of our paper being the conventional joint entity and relation extraction. To supplement the out-domain experiments, we conduct experiments in the dataset CrossRE [3] which is a cross-domain dataset for relation extraction. The dataset is available in [4]. There are six domains in CrossRE, namely news, politics, science, music, literature, and ai.
>
> Firstly, we apply OVO to the baseline in [3] to evaluate the effectiveness of our proposed method. The task of baseline is in-domain Relation Classification which needs to predict the relation type of entity pairs having a relation. As shown in (Supplement Table B1), OVO gains improvements in all domains of CrossRE with an average improvement of 0.8% and a best improvement of 1.6% in the news domain.
>
> (Supplement Table B1): Results in CrossRE baseline. $w.$ and $w/o.$ mean using OVO or not.
> | MICRO-F1(%)| news | politics | science | music | literature | ai   | avg  |
> | -------- | ---- | -------- | ------- | ----- | ---------- | ---- | ---- |
> | w.o.     | 46.4 | 58.3     | 40.1    | 76.0  | 67.7       | 45.4 | 55.7 |
> | w.       | 48.0 | 58.8     | 40.5    | 77.1  | 68.5       | 46.2 | 56.5 |
>
> Then, due to the missing Relation Extraction results in [3], we conduct experiments in CrossRE dataset using PL-Marker [5] in two out-domain settings:
> 	1) Overall out-domain setting: using the train set of one domain during training, using the test sets of the other five domains during testing.
> 	2) Fine-grained out-domain setting: using the train set of one domain during training, using the test set of another different domain during testing.
>
> In the overall out-domain setting, as shown in (Supplement Table B2), using OVO gains improvements in all cases ith an average improvement of 1.5% and a best improvement of 2.4% in the ai domain.
>
> (Supplement Table B2): Results in overall out-domain setting. $w.$ and $w/o.$ mean using OVO or not.
> |  Rel(%)  | news | science | politics | literature | ai   | music | avg  |
> | ---- | ---- | ------- | -------- | ---------- | ---- | ----- | ---- |
> | w/o. | 9.7  | 13.0    | 14.4     | 14.8       | 11.6 | 14.1  | 12.9 |
> | w.   | 10.3 | 14.3    | 15.4     | 16.4       | 14.0 | 16.4  | 14.5 |
>
> In the fine-grained out-domain setting, as shown in (Supplement Table B3), due to the limitation of time, we only conduct experiments in the news and science domains. The results show the out-domain effectiveness of OVO.
>
> (Supplement Table B3): Results in fine-grained out-domain setting. All values in the table are Rel(%). $w.$ and $w/o.$ mean using OVO or not; | means the same as above.
> | train | test       | w.o  | w.   | train   | test       | w.o  | w.   |
> | ----- | ---------- | ---- | ---- | ------- | ---------- | ---- | ---- |
> | news  | ai         | 10.1 | 10.3 | science | news       | 22.9 | 24.6 |
> | \|    | literature | 7.5  | 8.8  | \|      | ai         | 15.6 | 22.3 |
> | \|    | music      | 6.2  | 7.1  | \|      | literature | 10.2 | 11.3 |
> | \|    | science    | 9.6  | 11.8 | \|      | music      | 9.7  | 11.4 |
> | \|    | polotics   | 14.2 | 16.7 | \|      | polotics   | 16.6 | 18.2 |
>
> The results in (Supplement Table B2) and (Supplement Table B3) show the out-domain effectiveness of OVO. We will provide supplement experiments if we have a chance to upload the next version of our paper. Your feedback is greatly appreciated, and we are committed to enhancing the quality of our research.
>
> **[references]**
>
> [1] Yiquan W, Kun K, Yating Z, Xiaozhong L, Changlong S, Jun X, Yueting Z, Luo S, Fei W, et al. De-Biased Court’s View Generation with Causality[J], Computer Science, 2020, 2020.emnlp-main: 763-780.
>
> [2] Xiaobao Wu, Chunping Li, Yishu Miao. Discovering Topics in Long-tailed Corpora with Causal Intervention.[C], Annual Meeting of the Association for Computational Linguistics, 2021, 2021.findings-acl: 175-185.
>
> [3] Elisa Bassignana, Barbara Plank. CrossRE: A Cross-Domain Dataset for Relation Extraction[J], EMNLP (Findings), 2022: 3592-3604.
>
> [4] https://github.com/mainlp/CrossRE
>
> [5] Deming Y, Yankai L, Peng L, Maosong S, et al. Packed Levitated Marker for Entity and Relation Extraction[C], Annual Meeting of the Association for Computational Linguistics, 2022, Proceedings of the 60th Annual Meeting of the Association for Computational Linguistics (Volume 1: Long Papers): 4904–4917.

---

### Official Review · Reviewer_J2YN · 2023-08-10

**Soundness:** 3

**Excitement:**

3: Ambivalent: It has merits (e.g., it reports state-of-the-art results, the idea is nice), but there are key weaknesses (e.g., it describes incremental work), and it can significantly benefit from another round of revision. However, I won't object to accepting it if my co-reviewers champion it.

**Paper Topic And Main Contributions:**

The paper tried to solve the problem of distribution bias in joint entity and relation extraction, such as long-tail and out-of-distribution, from a causal perspective. One method of debiasing (covariance optimizing) optimizes the feature covariance and adjusts the sample weights to eliminate collinearity and correlation; the other method (variance optimizing) is based on backdoor adjustment to minimize the within-class variance. The authors conducted experiments on joint entity and relation extraction task and achieved significant improvement in several datasets.

**Reasons To Accept:**

1. The experiments are well-designed and are, in my opinion, sufficient to support the points.
2. The proposed method achieves significant improvements in several datasets of joint relation and entity extraction task.

**Reasons To Reject:**

1. In variance optimizing, the proposed SCM is not detailed enough. Why Real Label -> Prediction and why Real Label -> Feature vector? In Line 331-335 and Appendix A, the authors roughly state that it is because of model training, lacking in-depth discussion. This does not convince me that SCM is correct. It is worth noting that [1] attempts to address the long-tail problem from the perspective of optimizer momentum. I think the authors also need to provide an in-depth theoretical analysis of these causes and effects reflected in model training.
2. In the covariance optimizing, the authors did not make it clear why the proposed method can reduce the effect of collinearity and correlation. Like above, more analysis is needed.
3. I think some related work on causal inference needs to be reviewed. It is expected that more works related to the paper would be discussed, e.g., [1] [2] [3].
4.  lack of analysis of different batch sizes, which is important for MoCo training.
5. The proposed method should not only be applied to information extraction, I think the authors need to analyze the theory behind the method from a more general perspective. And it is suggested to study the effectiveness of the method from a previous setting of causal representation learning.

[1] Long-Tailed Classification by Keeping the Good and Removing the Bad Momentum Causal Effect

[2] Uncovering Main Causalities for Long-tailed Information Extraction

[3] De-biasing Distantly Supervised Named Entity Recognition via Causal Intervention

**Reproducibility:**

4: Could mostly reproduce the results, but there may be some variation because of sample variance or minor variations in their interpretation of the protocol or method.

**Reviewer Confidence:**

4: Quite sure. I tried to check the important points carefully. It's unlikely, though conceivable, that I missed something that should affect my ratings.

---

> ### Author Rebuttal · Authors · 2023-08-29
>
> We sincerely appreciate your thoughtful review of our paper. Your insightful comments and valuable feedback have significantly improved the quality of our work. In this response, we provide a detailed response to each of your points.
>
> 1. In the context of variance optimization, our primary objective is to address the bias stemming from the presence of long-tail distributions, which is induced by an imbalance in labels within datasets. We shall provide a comprehensive exposition of the causal graph as delineated below:
>
>      To simplify the description, we use the same abbreviations in the paper:
>
>      L : Real Label
>
>      F : Feature vector
>
>      P : Prediction
>
>     In our causal graph, P represents the outputs of the model, while L represents the real labels of samples. Throughout the training process, our primary objective is to minimize the disparity between P and L, a task typically quantified using a loss function, such as cross-entropy. Consequently, to capture and depict this relationship between L and P, we introduce an edge **L -> P** within the causal graph.
>
>    Moreover, within datasets characterized by a long-tail distribution of labels, it is observed that labels associated with a higher frequency of occurrence tend to receive more frequent predictions from the model. This observation is a direct outcome of the model's learning process, wherein optimization is primarily directed toward enhancing accuracy with respect to the predominant class. Consequently, to represent the impact of label distribution on the feature vectors of data samples, we introduce an edge, symbolized as **L -> F**, into our causal graph.
>
>    In order to ameliorate the aforementioned phenomenon, we propose the incorporation of a novel optimization target during the training phase. By minimizing the variance in the feature vectors of each label, as shown in Table 4, the performances of the labels with a fewer number of samples gain significant improvement.
>
>
>
> 2. Collinearity and correlation phenomena manifest as a consequence of interrelationships between variables. In the context of covariance optimization, we employ the covariance metric to quantitatively assess the degree of correlation existing between variables. In COP, our primary aim is to mitigate this correlation. We posit that a reduction in covariance signifies a diminishment of shared associations among variables, consequently leading to a decrease in both collinearity and correlation among these variables.
>
>
>
> 3. We appreciate your suggestion and intend to conduct a comprehensive review of the pertinent literature pertaining to causal inference.
>
>
>
> 4. The concern in question was discussed within our paper. Specifically, documented in Appendix D.1, we evaluate the impact of saving batch numbers.
>
>
>
> 5. Thank you for your valuable feedback. We appreciate your suggestion to analyze the proposed method from a more general perspective and to explore its effectiveness in the context of causal representation learning. While we acknowledge the importance of these directions, our current research is focused on joint entity and relation extraction tasks. It's indeed challenging to cover all possible applications and theoretical aspects in a single study.
>
> **[5 continued]**
>
> To further evaluate the applicability of OVO, we conduct supplement experiments during rebuttal.
>
> - 1) We apply OVO to the baseline of CrossRE [1]. The task is Relation Classification which needs to predict the relation type of entity pairs having a relation. The results is reported in (Supplement Table A1).
>
> - 2) To evaluate the effectiveness of the out-domain of OVO, we conduct out-domain experiments using CrossRE.
>   - Firstly, we evaluate in overall out-domain setting which uses the train set of one domain during training, using the test sets of the other all domains during testing.
>   - Secondly, we evaluate in fine-grained out-domain setting which uses the train set of one domain during training, using the test set of another different domain during testing.
>
> The results of the overall out-domain setting and the fine-grained out-domain setting are shown in (Supplement Table A2) and (Supplement Table A3) respectively.
>
> (Supplement Table A1): Results in CrossRE baseline. $w.$ and $w/o.$ mean using OVO or not.
>
> | MICRO-F1(%) | news | politics | science | music | literature | ai   | avg  |
> | ----------- | ---- | -------- | ------- | ----- | ---------- | ---- | ---- |
> | w.o.        | 46.4 | 58.3     | 40.1    | 76.0  | 67.7       | 45.4 | 55.7 |
> | w.          | 48.0 | 58.8     | 40.5    | 77.1  | 68.5       | 46.2 | 56.5 |
>
> (Supplement Table A2): Results in overall out-domain setting. $w.$ and $w/o.$ mean using OVO or not.
>
> | Rel(%) | news | science | politics | literature | ai   | music | avg  |
> | ------ | ---- | ------- | -------- | ---------- | ---- | ----- | ---- |
> | w/o.   | 9.7  | 13.0    | 14.4     | 14.8       | 11.6 | 14.1  | 12.9 |
> | w.     | 10.3 | 14.3    | 15.4     | 16.4       | 14.0 | 16.4  | 14.5 |
>
> (Supplement Table A3): Results in fine-grained out-domain setting. All values in the table are Rel(%). $w.$ and $w/o.$ mean using OVO or not; | means the same as above.
>
> | train | test       | w.o  | w.   | train   | test       | w.o  | w.   |
> | ----- | ---------- | ---- | ---- | ------- | ---------- | ---- | ---- |
> | news  | ai         | 10.1 | 10.3 | science | news       | 22.9 | 24.6 |
> | \|    | literature | 7.5  | 8.8  | \|      | ai         | 15.6 | 22.3 |
> | \|    | music      | 6.2  | 7.1  | \|      | literature | 10.2 | 11.3 |
> | \|    | science    | 9.6  | 11.8 | \|      | music      | 9.7  | 11.4 |
> | \|    | polotics   | 14.2 | 16.7 | \|      | polotics   | 16.6 | 18.2 |
>
> [references]
>
> [1] Elisa Bassignana, Barbara Plank. CrossRE: A Cross-Domain Dataset for Relation Extraction[J], EMNLP (Findings), 2022: 3592-3604.

---

### Meta-Review · Area_Chair_HHxQ · 2023-09-07

**Recommendation:** 3

**Metareview:**

This work introduces a covariance and variance optimization framework (OVO) to debias NER and RE tasks that is based on structural causal inference like a few other debiasing methods, but leads to more fine-grained adjustment than most of those. The technical novelty of this method is generally acknowledged by the reviewers. The paper originally missed important results on out-of-distribution evaluation, which is necessary for debiasing methods. There have also been a series of important comparative methods there were originally missed. The authors have provided some of those in the discussion phase, and the AC believe that these are essential to be included into the draft should the paper be accepted. In addition, there seems to be flaws identified regarding the SCM used behind the method, for which the discussion didn't reach consensus among the authors and the reviewers.

---

### Decision · Program_Chairs · 2023-10-07

**Decision:**

Accept-Findings

**Comment:**

This work introduces a covariance and variance optimization framework (OVO) to debias NER and RE tasks that is based on structural causal inference like a few other debiasing methods, but leads to more fine-grained adjustment than most of those. The technical novelty of this method is generally acknowledged by the reviewers. The paper originally missed important results on out-of-distribution evaluation, which is necessary for debiasing methods. There have also been a series of important comparative methods there were originally missed. The authors have provided some of those in the discussion phase, and the AC believe that these are essential to be included into the draft should the paper be accepted. In addition, there seems to be flaws identified regarding the SCM used behind the method, for which the discussion didn't reach consensus among the authors and the reviewers.